# Growth on stiffer substrates impacts animal health and longevity in *C. elegans*

**Maria Oorloff**[1◉], **Adam Hruby**[1◉], **Maxim Averbukh**[1], **Athena Alcala**[1], **Naibedya Dutta**[1], **Cray Minor**[2], **Toni Castro Torres**[1], **Darius Moaddeli**[1], **Matthew Vega**[1], **Juri Kim**[1], **Andrew Bong**[1], **Aeowynn J. Coakley**[1], **Daniel Hicks**[1], **Jing Wang**[1], **Tiffany Wang**[1], **Sally Hoang**[1], **Kevin M. Tharp**[2], **Gilberto Garcia**[1], **Ryo Higuchi-Sanabria**[1]*

1 Leonard Davis School of Gerontology, University of Southern California, Los Angeles, CA, United States of America, 2 Cancer Metabolism and Microenvironment Program, Sanford Burnham Prebys, La Jolla, CA, United States of America

◉ These authors contributed equally to this work.
* ryo.sanabria@usc.edu

**Data Availability Statement:** All data required to evaluate the conclusions in this manuscript are available within the paper and Supporting Information. All strains synthesized in this manuscript are derivatives of N2 or other strains

## Abstract

Mechanical stress is a measure of internal resistance exhibited by a body or material when external forces, such as compression, tension, bending, etc. are applied. The study of mechanical stress on health and aging is a continuously growing field, as major changes to the extracellular matrix and cell-to-cell adhesions can result in dramatic changes to tissue stiffness during aging and diseased conditions. For example, during normal aging, many tissues including the ovaries, skin, blood vessels, and heart exhibit increased stiffness, which can result in a significant reduction in function of that organ. As such, numerous model systems have recently emerged to study the impact of mechanical and physical stress on cell and tissue health, including cell-culture conditions with matrigels and other surfaces that alter substrate stiffness and ex vivo tissue models that can apply stress directly to organs like muscle or tendons. Here, we sought to develop a novel method in an in vivo model organism setting to study the impact of altering substrate stiffness on aging by changing the stiffness of solid agar medium used for growth of *C. elegans*. We found that greater substrate stiffness had limited effects on cellular health, gene expression, organismal health, stress resilience, and longevity. Overall, our study reveals that altering substrate stiffness of growth medium for *C. elegans* has only mild impact on animal health and longevity; however, these impacts were not nominal and open up important considerations for *C. elegans* biologists in standardizing agar medium choice for experimental assays.

## Introduction

Mechanical stress, defined as internal forces placed within a material or structure, can be classified into various types: tensile stress (stretching or pulling apart), compressive stress (squeezing or pushing together), shear stress (parallel forces acting in opposite directions), and torsional stress (twisting) [1]. In the context of living organisms, mechanical stress can have a dramatic impact on cell physiology, especially in tissues and organs subject to mechanical

from CGC and are either made available on CGC or available upon request. All raw RNA-seq datasets are available through Annotare 2.0 Array Express Accession E-MTAB-13938.

**Funding:** A.H., M.A., and G.G. are supported by T32AG052374; M.O., T.C.T., and M.V. are supported by 1R25AG076400 from the National Institute on Aging; T.W. is supported by the California Institute for Regenerative Medicine COMPASS Award EDUC5-13853; and R.H.S. is supported by R00AG065200 from the National Institute on Aging, Larry L. Hillblom Foundation Grant 2022-A-010-SUP, and the Glenn Foundation for Medical Research and AFAR Grant for Junior Faculty Award. Some strains were provided by the CGC, which is funded by the NIH Office of Research Infrastructure Programs (P40 OD010440). Some gene analysis was performed using Wormbase, which is funded on a U41 grant HG002223. The funders had no role in study design, data collection and analysis, decision to publish, or preparation of the manuscript" is correct.

**Competing interests:** The authors have declared that no competing interests exist.

forces such as the muscle, bone, and blood vessels. For example, tendon and muscle tissue contain highly mechanosensitive tissue that can remodel the extracellular matrix (ECM) in response to local loading environments, which is an important response to ensure proper development, maintenance, and repair of the tissue [2]. Even on the cellular level, different substrate stiffnesses can impact cell morphology, growth, proliferation, differentiation, and matrix remodeling [3].

In response to physical forces, cells and tissues convert these mechanical forces into biochemical activities. Termed mechanochemical transduction or mechanotransduction, these events generally include extracellular receptors that initiate a signaling cascade in the cell that can regulate gene expression. For example, a wide variety of integrin receptors have been discovered that can sense changes in the extracellular environment and relay these messages inside of the cell. Changes to the extracellular matrix, physical stimulation, and alterations in cell adhesions can all activate integrin subtypes that can result in dramatic remodeling inside of the cell [4]. This includes changes to ion balance, actin and mitochondrial remodeling, and phosphorylation events that can lead to activation of genes involved in stress response, organelle homeostasis, and ECM remodeling, just to name a few [5].

Mechanical stress is not only restricted to external stimuli; internal changes to tissue stiffness, such as ECM remodeling, can also initiate mechanotransduction events. This is most apparent in aging tissue, which tend to get stiffer with age due to a build-up of collagen that can contribute to tissue fibrosis [6]. This increase in tissue stiffness can have direct consequences on tissue function. For example, changes in collagen composition and cross-linking in the tendon with age can result in reduced tendon compliance, increased stiffness, and decreased resilience to mechanical stress, which is a direct cause of the age-related increase in tendon injuries [7, 8]. The aging ovary also displays a collagen-dependent increase in stiffness which results in improper follicle development and oocyte quality and an overall reduction in reproductive capacity. These changes are not only limited to collagen, but also involve changes to other components of the ECM, including a significant decrease in the glycosaminoglycan, hyaluronic acid (HA) [9]. In fact, HA is a commonly used anti-aging intervention for skin health as it can regulate skin moisture and elasticity [10], and even in cell culture models and model organisms, increased levels of HA can improve stress resilience and longevity [11]. Aging is not universally associated with an increase in stiffness, however. Human fibroblasts taken from donors under the age of 25 are significantly more stiff than those from donors over the age of 30 [12]. In addition, the stiffness of human dermal tissue was found to increase with age before decreasing after age 55 [13]. Notably, aged *C. elegans* exhibit a 10-fold decrease in stiffness as compared to young animals, and animals of the same age that are stiffer live longer, providing evidence that maintenance of mechanical integrity plays an important role in organismal aging [14].

Changes to ECM composition not only occur during aging but are a hallmark of cancer pathology. Although cancer cells themselves exhibit less stiffness than healthy cells [15], solid state tumors are characterized by a significant increase in stiffness due to increased density of collagen and other ECM components [16]. This change in tumor microenvironment can drive cancer progression, as increased adhesion-mediated mechanosignaling can initiate "pro-survival" mechanisms in cancer cells. Specifically, one study found that cancer cells growing on stiffer substrates activate an integrin-mediated remodeling of the cytoskeleton and mitochondria, which can activate cellular stress responses that improve cancer cell resilience [17]. This increase in stress resilience translated to increased drug-resistance, which can be ameliorated by downregulating these pathways, suggesting that targeting mechanotransduction pathways can have potential both in anti-aging and anti-cancer interventions.

Considering the significance of mechanotransduction in organismal health and aging, we sought to determine the effect of mechanical stress on an in vivo model system, the nematode worm *Caenorhabditis elegans*. Ideally, our goal was to identify the simplest method to apply mechanical stress onto this model organism in an effort to develop a model system that could be used to study how changes in mechanotransduction can impact the aging process. As such, we grew *C. elegans* on a stiffer substrate, which previous studies have shown can alter *C. elegans* locomotory behavior [18] by increasing the drag force against movement and thus requiring greater body bending force [19, 20] which we speculate induces mechanical stress. Here, we created a stiffer substrate by using double the concentration of agar (4%) compared to a standard solid agar medium (2%). This approach, while simple, is not without its limitations, such as an inability to control for differences in water content which will modify humidity and osmotic conditions. Although we expected this increase of substrate stiffness to have major impacts on organismal health, our work revealed that growth of animals on stiffer substrates did not activate canonical mechanotransduction pathways. Moreover, cellular health and transcriptional regulation were largely unaffected, although significant changes were seen in some metrics of physiology. Overall, while the increase in substrate stiffness did not increase mechanical stress as originally hypothesized, this study presents significant findings in how altering substrate stiffness can impact animal physiology.

## Materials and methods

### *C. elegans* maintenance

All strains used in this study are derivatives of the N2 wild-type strain from the Caenorhabditis Genetics Center (CGC) and a full strain list is available in **S3 Table**. All animals were grown at 15˚C on OP50 *E. coli* B strain and moved onto a new NGM plate seeded with OP50 every week for standard maintenance. To avoid genetic drift, every week > 20 individual animals were moved to new plates each passage for a maximum of 25–30 passages. All experimental strains were bleached to clarify OP50 bacteria and grown at 20˚C on HT115 E. *coli* K strain bacteria. For all experiments, animals were either grown on HT115 bacteria carrying a pL4440 plasmid that does not target any specific genes (called EV for empty vector in this manuscript) or pL4440 vector carrying a partial gene sequence against a specific target gene for RNAi-mediated knockdown.

**Plates.** Standard NGM plates for maintenance: Bacto-Agar (Difco) 2% w/v, Bacto-Peptone 0.25% w/v, NaCl 0.3% w/v, 1 mM $CaCl_2$, 5 µg/ml cholesterol, 0.625 mM $KH_2PO_4$ pH 6.0, 1 mM $MgSO_4$.

2% RNAi plates for experiments: Bacto-Agar (Difco) 2% w/v, Bacto-Peptone 0.25% w/v, NaCl 0.3% w/v, 1 mM $CaCl_2$, 5 µg/ml cholesterol, 0.625 mM $KH_2PO_4$ pH 6.0, 1 mM $MgSO_4$, 100 µg/mL carbenicillin, 1 mM IPTG.

4% RNAi plates for experiments: Bacto-Agar (Difco) 4% w/v, Bacto-Peptone 0.25% w/v, NaCl 0.3% w/v, 1 mM $CaCl_2$, 5 µg/ml cholesterol, 0.625 mM $KH_2PO_4$ pH 6.0, 1 mM $MgSO_4$, 100 µg/mL carbenicillin, 1 mM IPTG.

For all aging experiments, 100 µL of 10 mg/mL (+)-5-Fluorodeoxyuridine (FUDR) was placed directly on the bacterial lawn, unless otherwise stated.

**Bleaching.** All experiments were performed on synchronized, age-matched populations acquired through a standard bleaching protocol. Animals were collected into a 15 mL conical tube using M9 solution (22 mM $KH_2PO_4$ monobasic, 42.3 mM $Na_2HPO_4$, 85.6 mM NaCl, 1 mM $MgSO_4$). M9 solution was replaced with bleaching solution (1.8% sodium hypochlorite, 0.375 M NaOH, diluted in M9 solution) and animals are treated with bleaching solution until only eggs remain in the tube with no adult carcasses remaining (~4–6 minutes). Eggs were

then washed with M9 solution 4 times by centrifugation at 1.1 RCF for 30 seconds and aspiration of bleaching solution and replacing with M9 solution. Following the final wash, animals were L1 arrested in M9 solution in a rotator for up to 24 hours in 20°C.

## Microscopy

**Stereomicroscope.** For transcriptional reporter imaging (*hsp-6p*::*GFP*, *hsp-4p*::*GFP*, *hsp-16.2p*::*GFP*, *gst-4p*::*GFP*), synchronized animals were grown on either 2% RNAi or 4% RNAi plates from L1. Animals were imaged at day 1 of adulthood and for aging experiments also imaged at day 5 and day 9 of adulthood. Animals were placed on standard NGM plates without bacteria on 100 mM sodium azide to paralyze worms. Paralyzed animals were lined up alongside each other and imaged on a Leica M205FCA automated fluorescent stereomicroscope equipped with a standard GFP and RFP filter, Leica LED3 light source, Leica K5 camera, and run on LAS X software. Images were quantified with Fiji [25] by using the subtract background function with a rolling ball radius of 50.0 pixels, drawing a region of interest along each group of worms, and measuring integrated density.

**Widefield and confocal imaging.** For high resolution imaging of organelles including actin, mitochondria, ER, lysosomes, and lipid droplets, we performed imaging on a Leica THUNDER widefield microscope equipped with a 63x/1.4 Plan AproChromat objective, standard GFP and dsRed filter, Leica DFC9000 GT camera, a Leica LED5 light source, and run on LAS X software, or a Leica Stellaris 5 confocal microscope equipped with a white light laser source and spectral filters, HyD detectors, 63x/1.4 Plan ApoChromat objective, and run on LAS X software. Animals were placed in 100 mM sodium azide solution on a glass slide and imaged within 5 minutes of slide preparation to prevent artifacts from animals being on a glass slide in sodium azide for extended periods of time. For all mitochondrial morphology imaging, no sodium azide was used, and animals were placed on a glass slide with M9 solution and imaged within 5 minutes to prevent mitochondrial fragmentation.

## RNA sequencing analysis

RNA isolation was performed on day 1 of adulthood. ~1000 animals were grown on either 2% or 4% RNAi plates with empty vector (EV) RNAi bacteria and animals were harvested from plates using M9 solution. Animals were gravity settled to allow adult worms to sink to the bottom and eggs and L1 larvae were aspirated off the top. Animals were washed 3 times in this manner to minimize the number of progeny collected. Animals were then placed into Trizol solution and freeze/thawed 3 times with cycles of liquid nitrogen and 37°C bead bath for 1 min. Before every freeze/thaw cycle, animals were vortexed for 30 seconds to ensure complete digestion of animal cells. After the final thaw, chloroform was added at a 1:5 chloroform:trizol ratio and aqueous separation of RNA was performed via centrifugation in a heavy gel phase-lock tube (VWR, 10847–802). The aqueous phase was mixed 1:1 with isopropanol then applied to a Qiagen RNeasy Mini Kit (74106), and RNA purification was performed as per manufacturer's directions. Library preparation was performed using a Kapa Biosystems mRNA Hyper Prep Kit. Sequencing was performed at the Vincent J Coates Genomic Sequencing Core at the University of California, Berkeley using an Illumina HS4000 mode SR100. Three biological replicates were measured per condition. Reads per gene were quantified using kallisto [26], with WBcel235 as the worm reference genome. Fold changes were determined using DESeq2 [27]. All raw RNA-seq datasets are available through Annotare 2.0 Array Express Accession E-MTAB-13938.

## Lifespan measurements

For lifespan experiments, animals were grown on either 2% or 4% agar RNAi plates on either empty vector (EV) or RNAi bacteria from L1 (see bleaching). Animals were then moved to their respective plates containing 100 µL of 100 mg/mL FUDR starting from day 1 of adulthood to eliminate progeny. For stress assays, animals were moved onto plates supplemented with either 2.5 mM paraquat or 25 µg/mL tunicamycin directly in the plate. Animals were grown at 20˚C and checked every 2 days for viability. For thermotolerance, day 1 adult animals were placed at 34˚C and scored for viability every 2 hours. Animals were scored as dead if they did not exhibit any movement when prodded with a platinum wire at both the head and the tail. Animals that exhibited intestinal leakage, bagging, desiccation on the side of the petri dish, or other deaths unrelated to aging were scored as censored. A table of lifespans is available in **S1 Table**.

For dead bacteria assays, NGM agar plates were spotted with 200 µL of EV RNAi bacteria and allowed to dry overnight. Spotted plates were then treated in a UV cross-linker (CL-1000 Ultraviolet Crosslinker, 254 nm; Energy x 100 µ J/cm$^2$) for ten minutes. Both the spotted plate and lid were exposed to UV treatment face up to kill bacteria and maintain sterility. Treated plates were left at room temperature overnight prior to use to ensure that any heating of the plate from UV exposure did not affect animal health. Adult animals were moved away from progeny daily onto a fresh plate without the usage of FUDR for this assay.

## *C. elegans* motility

For all motility experiments, thrashing of animals was scored on day 1, day 5, and day 9 of adulthood with animals grown on either 2% or 4% RNAi plates from L1. Animals were collected using M9 solution and placed onto an NGM plate without bacteria and 10 second video recordings were taken using a Leica M205FCA stereomicroscope. Thrashing was manually recorded wherein a single body bend in one direction is scored as a single thrash and the total number of thrashes for a 10 second period are calculated.

## *C. elegans* reproduction assay

For all reproduction experiments, animals were grown on either 2% or 4% RNAi plates from L1 (see bleaching). 10 animals were singled out at the L4 stage and moved daily onto a new plate. The number of viable progeny on each plate was scored every day and summed per animal to calculate the number of viable progeny produced by each animal.

## Measurement of surface stiffness

For **Fig 5**, viscoelastic properties of NGM plates using agar from different commercially available sources were determined using an oscillatory rheometer (Anton-Paar, M302e) with parallel-plate geometry (8 mm) and a gap height of ∼ 0.2 mm under 0.1% strain and 1 Hz frequency at 37˚C in a humidity-controlled chamber. For **S5 Fig**, viscoelastic properties of agar gels were determined using an oscillatory rheometer (Anton-Paar, MCR102e) with parallel-plate geometry (8mm), with normal force ~0.1N. Frequency sweeps were performed in the range of 1–100 rad/s (0.159–15.9 Hz) under constant 1% strain at room temperature (~21˚C) to determine storage (G′) and loss (G″) moduli, to describe the deformation behavior of the samples in the non-destructive linear-viscoelastic (LVE) range.

## Statistics

All statistical analyses were performed using GraphPad Prism 10 software. No assumptions were made about data distribution. For motility, reproduction, and stress reporter assays,

Mann-Whitney testing was used with p-values less than 0.05 considered significant. To determine statistical significance of lifespan data, a log-rank test was performed with p-values less than 0.05 considered significant. At least 3 biological replicates were performed for each experiment unless otherwise noted.

## Results and discussion

To develop a mechanical stress model for *C. elegans*, here we sought to alter substrate stiffness by growing animals on higher percentages of agar. Our standard lab-growth nematode growth medium (NGM) contains 2% agar w/v and for the entirety of this study, we used Bacto Agar Solidifying Agent from BD Diagnostics (catalog no. 281210). For our stiffer substrate, we utilized 4% agar w/v, as this was the highest concentration of agar we could use without having issues with the agar either precipitating or solidifying prior to cooling sufficiently to add liquid additives for standard NGM plate-pouring (refer to [28] for a detailed protocol on making NGM plates). We did not include a softer substrate in this study as we found that growth on 1% agar resulted in a majority of animals burrowing into the agar medium, making experimentation and drawing conclusions from the data challenging. First, we confirmed that 4% agar plates had a significantly higher (approximately 4x) substrate stiffness when compared to 2% agar plates (**S1 Fig**). When wild-type animals were grown on stiffer substrates from the L1 larval stage, we found that animals exhibited a mild, but significant increase in lifespan (**Fig 1A**). This lifespan extension was not due to changes in bacterial physiology or the usage of FUDR, as growth on 4% agar still increased lifespan on UV-treated bacteria with no FUDR treatment (**Fig 1B**). Interestingly, although animals live slightly longer, they display a mild decrease in specific healthspan metrics, including locomotory behavior measured by a liquid thrashing assay (**Fig 1C**) and reproductive capacity (**Fig 1D**). Therefore, although animals are slightly longer-lived when grown on stiffer substrates, there appears to be a tradeoff in healthspan.

Here, we reasoned that growth on stiffer substrates may extend lifespan by activation of specific stress response pathways. Numerous previous studies have shown hyperactivation of specific stress responses, including the unfolded protein responses of the mitochondria (UPR$^{MT}$) [29], the endoplasmic reticulum (UPR$^{ER}$) [30], and heat-shock response (HSR) [31] all extend longevity. In addition, several of these stress response-mediated longevity paradigms are associated with a decrease in reproductive potential [28, 32], which mirrors phenotypes found on animals grown on stiffer substrates. However, we found that stress responses were largely unchanged (**Fig 2**). Specifically, we measured activation of stress responses using transcriptional reporters whereby GFP expression is under the promoter of canonical genes activated upon stress. For the UPR$^{MT}$, the promoter of the mitochondrial chaperone, *hsp-6* (HSPA9 in mammals) [33] is used. Growth on 4% agar had no impact on *hsp-6p::GFP* expression, both in the presence (RNAi knockdown of the electron transport chain component, *cco-1* [29]) and absence of stress (**Fig 2A**). Similarly, we measured UPR$^{ER}$ induction using the *hsp-4p::GFP* (HSPA5/BiP) reporter [34] and found no activation of the UPR$^{ER}$ upon growth on stiffer substrates. Upon imaging, there appeared to be a mild increase in the UPR$^{ER}$ expression under stress applied by RNAi knockdown of *tag-335* involved in N-linked glycosylation of proteins in the ER, knockdown of which drives accumulation of misfolded proteins in the ER. However, quantification revealed that there were no significant differences between 2% and 4% UPR$^{ER}$ induction under stress. Similarly, there were no measurable changes in HSR using the *hsp-16.2p::GFP* (CRYAB) reporter [35] or oxidative stress response using the *gst-4p::GFP* (HPGDS) reporter [36] both in the presence (2-hour heat shock at 34°C or tert-butyl hydroperoxide exposure) and absence of stress (**Fig 2C and 2D**).

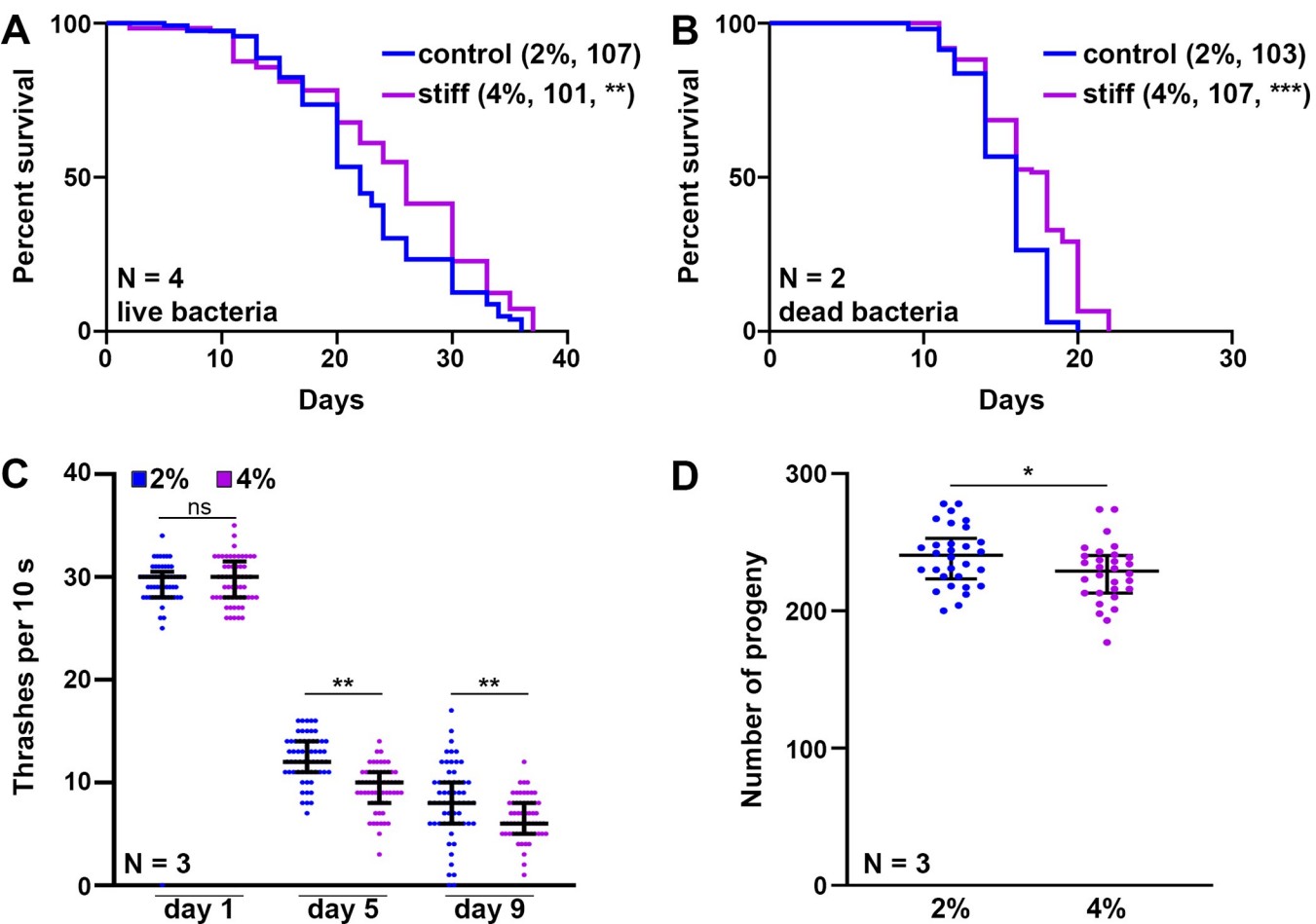

**Fig 1. Growth on stiff substrate results in mild lifespan extension but a decrease in organismal health.** (A) N2 wild-type animals grown on empty vector (EV) RNAi bacteria on either control (2%, blue) or stiff (4%, purple) agar plates from L1. Lifespans were scored every 2 days. Data is representative of 4 biological replicates, sample size is represented in the legend in parentheses, and statistical analysis is available in **S2 Table**. ** p = 0.0031. **(B)** N2 wild-type animals grown on EV RNAi bacteria on either control (2%, blue) or stiff (4%, purple) agar plates from L1. Animals are grown on UV-treated bacteria with no FUDR. Lifespans were scored every 2 days. Data is representative of 2 biological replicates, sample size is represented in legend in parentheses, and statistical analysis is available in **S2 Table**. *** p < 0.0001. **(C)** N2 wild-type animals were grown on EV RNAi bacteria on either 2% or 4% NGM plates from L1. At day 1, 5, and 9 of adulthood, animals were collected in M9 solution and video recordings were taken on an M205 stereoscope for 10 seconds. Body bends per 10 seconds were counted by eye for each individual worm. **(D)** N2 wild-type animals were grown on EV RNAi bacteria on either 2% or 4% agar plates from L1. At the L4 stage, animals were singled out and moved daily. Number of live progenies were counted for each animal with n > 50 per sample. Data is pooled from 3 independent biological replicates with n = 10 per replicate. * = p < 0.05, ** = p < 0.01 through non-parametric Mann-Whitney testing. Lines are median and interquartile range, and each dot represents a single animal where blue dots are animals grown on 2% control plates and purple dots are animals grown on 4% stiff plates.

Although transcriptional reporters provide a robust and rapid method of screening for activation of stress responses, a major limitation is that they focus on a single target, whereas stress responses tend to involve major and dramatic remodeling of transcription. Therefore, to obtain a more comprehensive overview of transcriptional changes driven upon growth on stiffer substrates, we performed bulk RNA-seq analysis comparing animals grown on 2% and 4% agar plates (**Fig 3**). We saw very minor changes in transcription, with only 35 genes differentially expressed (with an adjusted p < 0.05 cutoff) in animals grown on stiffer substrates (**Fig 3A and 3B**). Importantly, very few of these gene expression changes were those found in canonical mechanical stress or mechanotransduction pathways, such as remodeling of the ECM, mitochondria, or actin cytoskeleton. Only two GO-terms associated with extracellular

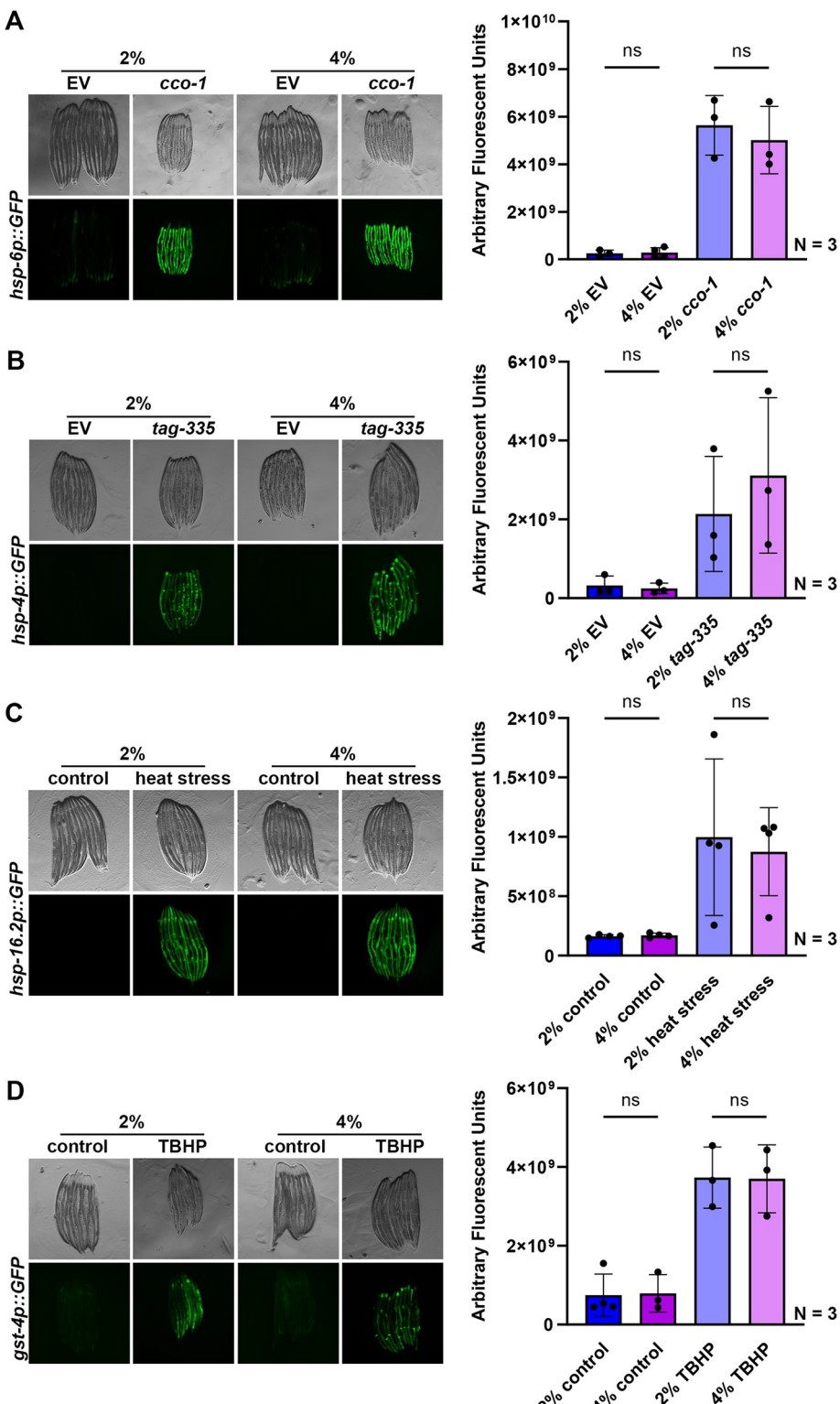

**Fig 2. Growth on stiff substrates does not affect induction of stress responses. (A)** Representative fluorescent images of day 1 adult animals expressing *hsp-6p*::*GFP* grown on EV or *cco-1* RNAi bacteria from L1. Data is representative of 3 independent replicates. Quantification is presented as arbitrary fluorescent units, which are integrated fluorescent intensity measurements using ImageJ Fiji, dots are independent replicates, and lines represent standard deviation. Data is representative of 3 independent replicates. ns = not significant, p > 0.05. **(B)** Representative

fluorescent images of day 1 adult animals expressing *hsp-4p::GFP* grown on EV or *tag-335* RNAi bacteria from L1. Data is representative of 3 independent replicates. Quantification is presented as arbitrary fluorescent units, which are integrated fluorescent intensity measurements using ImageJ Fiji, dots are independent replicates, and lines represent standard deviation. Data is representative of 3 independent replicates. ns = not significant, p > 0.05. **(C)** Representative fluorescent images of day 1 adult animals expressing *hsp-16.2p::GFP* grown on EV RNAi bacteria from L1. Animals were heat-shocked at 34˚C for 2 hours followed by a 2-hour recovery at 20˚C. Data is representative of 3 independent replicates. Quantification of is presented as arbitrary fluorescent units, which are integrated fluorescent intensity measurements using ImageJ Fiji, dots are independent replicates, and lines represent standard deviation. Data is representative of 3 independent replicates. ns = not significant, p > 0.05. **(D)** Representative fluorescent images of day 1 adult animals expressing *gst-4p::GFP* grown on EV RNAi bacteria from L1. Animals were treated with 1 mM tert-butyl hydroperoxide (TBHP) rotating in an M9-TBHP solution for 2 hours at 20˚C. TBHP was then washed with M9 solution and worms were recovered on OP50 bacteria for 16 hours prior to imaging. Data is representative of 3 independent replicates. Quantification is presented as arbitrary fluorescent units, which are integrated fluorescent intensity measurements using ImageJ Fiji, dots are independent replicates, and lines represent standard deviation. Data is representative of 3 independent replicates. ns = not significant, p > 0.05.

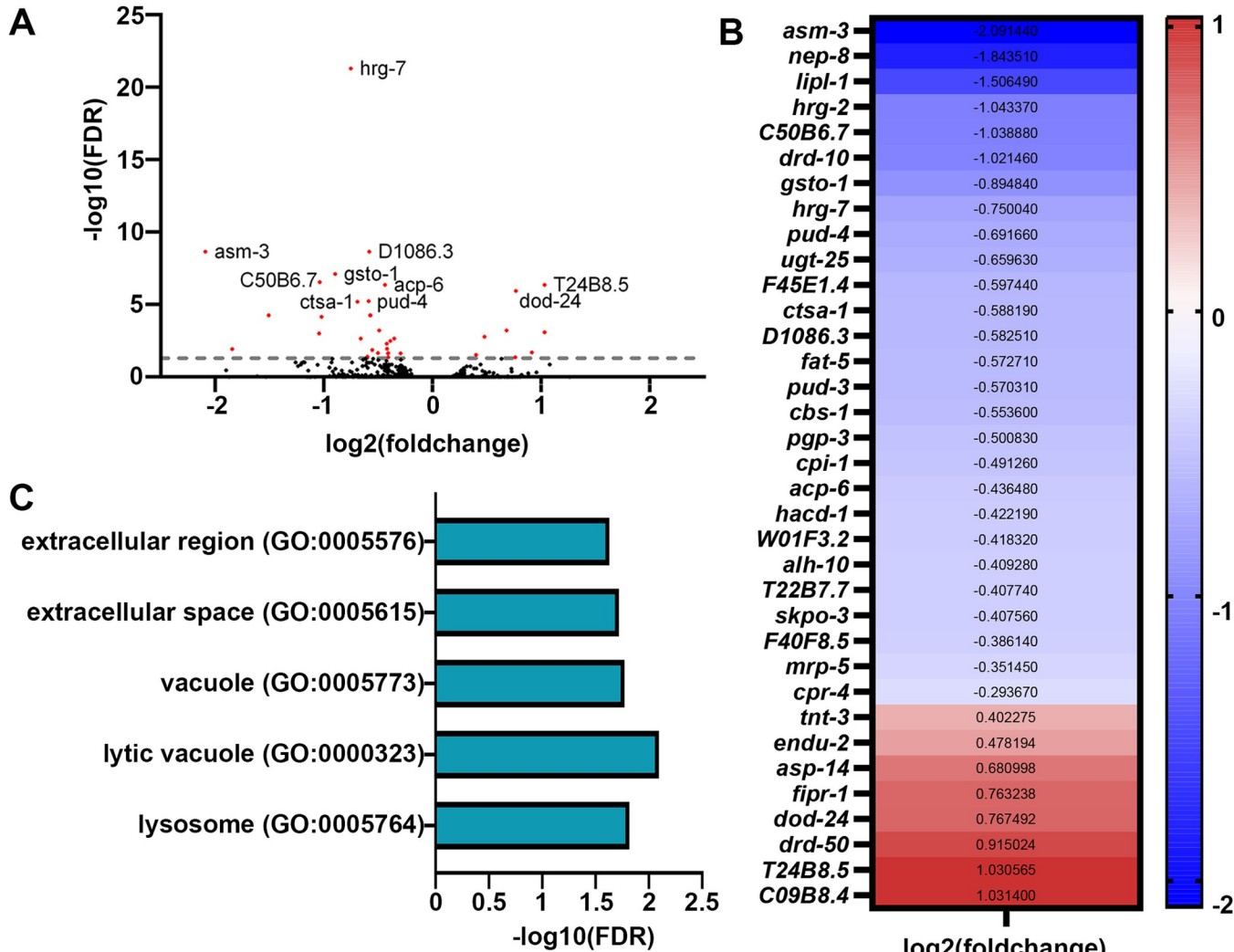

**Fig 3. Transcriptome analysis reveals minor changes in gene expression in animals grown on stiffer substrates. (A)** Volcano plot of differentially expressed genes of animals grown on stiff (4%) agar compared to standard (2%) agar. Every dot is a single gene, where red dots indicate genes with p-value < 0.05 and black dots indicate genes with p-value > 0.05. **(B)** Heat map of all differentially expressed genes in worms grown on 4% agar where warmer colors indicate higher expression and cooler colors indicate lower expression. A list of all genes is available in **S2 Table**. **(C)** Gene ontology enrichments for differentially expressed genes (p-value < 0.05) in worms grown on 4%'agar.

areas of the cell were identified (**Fig 3C**). Instead, the primary change we saw were those associated with the vacuole and lysosome, including *lipl-1*, encoding a key lysosomal lipase involved in lipophagy [37], and *asm-3*, encoding a sphingomyelinase involved in sphingomyelin breakdown, potentially through the lysosome [38]. To determine whether these changes in expression of lysosome and lipid-recycling related genes had direct physiological impact, we measured lysosome and lipid droplet content. However, we did not observe any major differences in lysosomal quantity using LMP-1::GFP [39] (**S2A Fig**), nor did we see any changes in lipid droplets as measured using DHS-3::GFP [40] (**S2B and S2C Fig**).

Since we saw a slight visible–although not statistically significant–increase in UPR$^{ER}$ activation under conditions of stress in animals grown on stiffer substrates, we next sought to determine whether this potentially mild change had a physiological impact. To test this, we grew animals on tunicamycin, which induces ER stress and significantly decreases lifespan [30]. Interestingly, we find that animals grown on stiffer substrates have a significant increase in survival under exposure to tunicamycin, which was dependent on the main UPR$^{ER}$ transcription factor XBP-1, as knockdown of *xbp-1* suppressed this increase in resilience (**S3A Fig**). Importantly, animals can activate UPR$^{ER}$ to the same extent throughout aging on 4% agar + tunicamycin plates, confirming that this difference in lifespan is not through altered efficiency of the drug in 4% plates (**S3C and S3D Fig**). Together, these data suggest that animals grown on stiffer substrates likely exhibit a functional increase in UPR$^{ER}$ activity upon stress exposure, which is dependent on canonical induction of UPR$^{ER}$ through XBP-1 [41]. Importantly, this is specific to ER stress resilience as animals on stiffer substrates did not exhibit an increase in oxidative or mitochondrial stress as measured by survival upon exposure to paraquat (**S3B Fig**), which induces superoxide formation in the mitochondria [42].

Previous studies have shown that exposure to ER stress can result in dramatic remodeling of the ER and activation of lipophagy machinery [23]. Since we saw an increase in ER stress resilience, we next visualized ER morphology using an mRuby::HDEL fusion protein that localizes to the ER lumen through a SEL-1 signal sequence [23]. However, we did not see any major differences in ER morphology (**S4A Fig**).

To get a better sense of wholescale cellular changes upon growth on stiffer substrates, we expanded our studies to changes in mitochondria and actin organization. Previous research has shown that growth on stiffer substrates results in integrin-dependent remodeling of the mitochondria and actin cytoskeleton in human cells [17]. Notably, integrin-mediated signaling also affects mitochondria and actin remodeling in *C. elegans* [43]. Thus, we were interested to know whether growth on stiffer substrates could impact mitochondria and actin in whole worms. Here, we visualized mitochondrial morphology in the muscle, intestine, and hypodermis of *C. elegans* using a mitochondria-targeted GFP [21] and cell-type specific promoters (**S4B and S4D Fig**). We find that mitochondria were fragmented in the muscle, intestine, and hypodermis in animals grown on stiffer substrates. To measure actin organization, we used the F-actin binding protein, LifeAct::mRuby, expressed specifically in the muscle, intestine, and hypodermis [24]. As previously reported, the actin cytoskeleton shows marked deterioration during aging, visualized by decreased linear organization of actin filaments in the muscle [24]. Interestingly, we observed improved actin organization at advanced age in muscle cells of animals grown on stiffer substrates (**Fig 4A**). Although we can only speculate whether this is mediated through integrin signaling, these data are consistent with previous studies that showed that increased integrin signaling can prevent age-related decline in actin organization, which can also drive mitochondrial fragmentation [43].

The hypodermis displays rearrangement of the actin cytoskeleton to star-like structures that are likely endocytic vesicles at around day 3 to 5 of adulthood, which start to dissociate after day 7 [24]. Similar to muscle actin, growth on stiffer substrates results in a mild

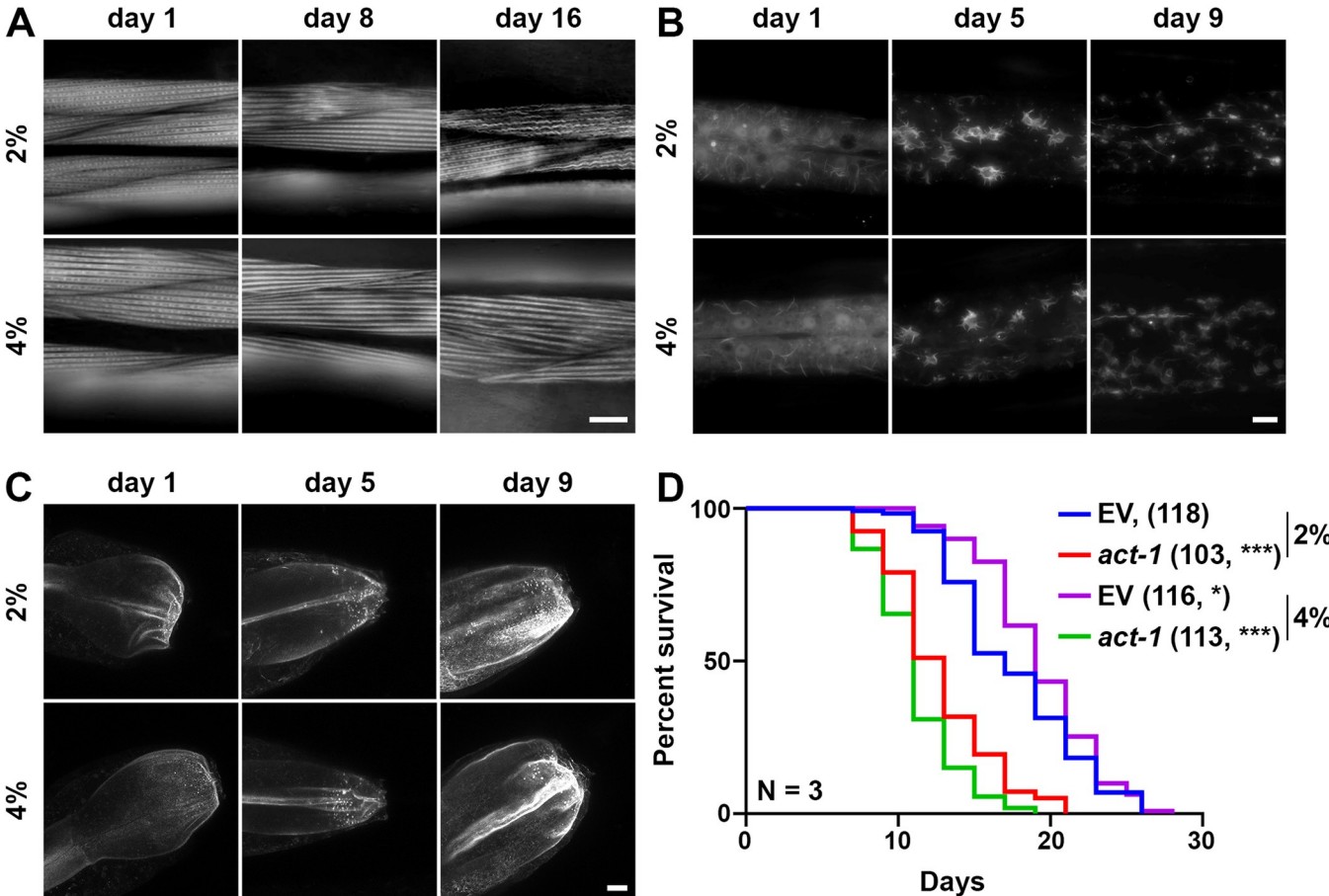

**Fig 4. Growth on stiff surfaces results in increased actin stability, which can drive lifespan extension.** (A) Representative fluorescent images of body wall muscle actin (*myo-3p*::*LifeAct*::*mRuby*) are shown. Animals were grown on empty vector (EV) RNAi bacteria from L1. Single-slice images were captured on a Leica THUNDER Imager. Scale bar is 10 μm. (B) Representative max-projection fluorescent images of hypodermal actin (*col-19p*::*LifeAct*::*mRuby*) are shown. Animals were grown on EV RNAi bacteria from L1. Z-stack images were captured on a Leica THUNDER Imager using system-optimized z-slices. Scale bar is 10 μm. (C) Representative max-projection fluorescent images of intestinal actin (*gly-19p*::*LifeAct*::*mRuby*) are shown. Animals were grown on EV RNAi bacteria from L1. Z-stack images were captured on a Leica Stellaris using system-optimized z-slices. Scale bar is 10 μm. (D) N2 wild-type animals grown on EV or *act-1* RNAi bacteria diluted 10/90 *act-1*/EV on either control (2%) or stiff (4%) agar plates from L1. Lifespans were scored every 2 days. Data is representative of 3 biological replicates, sample size is represented in the legend in parentheses, and statistical analysis is available in **S1 Table**. * p = 0.0012, *** p < 0.0001.

improvement in actin organization, such that more structures are preserved at late age (**Fig 4B**). Finally, actin in the intestine exhibits mislocalization and aggregation, which can result in decreased gut barrier function during aging [44]. Interestingly, we do not see improvements of these phenotypes in the intestine of animals grown on stiffer substrates (**Fig 4C**). To determine whether the mild increase in lifespan of animals grown on stiffer substrates is associated with the corresponding improvement in actin organization, we disrupted actin function using a carefully titrated, non-lethal dosage of actin RNAi as previously described [22]. Remarkably, we found that low-dose actin RNAi completely suppressed the lifespan extension found in these animals (**Fig 4D**). In addition, animals grown on stiff substrates displayed a slight increase in maximum thermotolerance (**S5A–S5C Fig**). Although in general, median thermotolerance did not show a significant increase, each replicate did display a significant difference from the 2% control (see **S2 Table** for statistical analysis). Importantly, this change in thermotolerance is not due to differences in plate composition as 4% plates heated up to 34°C in less than an hour, similar to 2% plates and the only difference between

the two plates was observable at 15 min and no other timepoint (**S5D Fig**). An increase in thermotolerance has previously been ascribed to an increase in cytoskeletal stability [45]. Altogether, these data show that growth on stiffer substrates results in improved actin organization in several tissue, which may underlie the mild increase in longevity and resilience to heat stress found in these animals.

## Conclusions

Mechanotransduction represents the complex interplay between the extracellular environment and the internal biological system. All living organisms are exposed to environmental changes that can apply mechanical forces, which are then converted into biochemical activities through a signaling cascade involving extracellular receptors and intracellular signaling pathways, culminating in remodeling of numerous intracellular components on the gene, protein, and organelle levels. For example, in human breast cancer cell lines, growth on stiffer substrates results in alterations of mitochondrial composition, structure, and function. These changes are driven by integrin-mediated mechanotransduction signals that alter solute transporters, including SLC9A1, which results in transcriptional activation of HSF1 to alter actin and mitochondrial function and oxidative stress response [17]. These dramatic intracellular changes both on the transcriptional and organellar level occur in response solely to alterations in substrate stiffness and have direct ramifications for cancer cell survival.

First, we tested the impact of growing whole animals on stiffer substrates. Specifically, the nematode *C. elegans* was grown on NGM containing 4% agar compared to the standard 2% agar concentration, which results in a stiffer plate. While we aimed to induce mechanical stress through increased substrate stiffness, it's important to recognize that increasing agar content introduces a number of variables that can impact worm physiology. Increased agar content necessarily reduces water content which can impact humidity and osmolarity, to which *C. elegans* is sensitive [46, 47]. Stiffer substrates also affect the physiology of bacteria [48]; thus, it's conceivable that changes to the bacterial food source may have an impact on *C. elegans*. However, growth on UV-treated bacteria still elicited the increase in lifespan observed on 4% agar plates, suggesting this was not a significant factor. Regardless, we cannot rule out that the phenotypes observed in this study were at least in part due to these confounding factors. Nevertheless, our results recapitulate a handful of phenotypes found in in vitro studies of mechanical stress.

First, animals exhibited improved actin organization with age, which is consistent with previous studies that showed that increased integrin signaling in *C. elegans* results in increased stability of actin filaments at late age [43]. Moreover, this improvement in actin organization correlates with increased longevity, again consistent with several studies that have linked actin function and organization to lifespan in *C. elegans* [22, 24]. Despite this actin-dependent increase in lifespan, animals grown on stiffer substrates did not exhibit major changes in gene expression of actin-related genes. It is possible that regulation of actin downstream of exposure to stiffer substrates is not dependent on transcriptional changes, but rather to protein signaling events, such as those mediated by Rho GTPases, an important actin-regulatory signaling switch downstream of mechanosensing [49]. Indeed, actin remodeling downstream of increased integrin signaling in human cells is dependent on RhoA/RAC signaling [17]. However, the lack of a general mechanotransduction gene expression signature suggests that perhaps whole animals grown on stiffer substrates result in activation of different signaling pathways than cells grown directly on stiffer substrates. For example, it is possible that the thick cuticle of *C. elegans*–the part of the whole animal that interacts with the agar substrate–has dramatic differences in mechanosensing and downstream mechanotransduction

compared to cells. Alternatively, the maintenance of body shape in spite of the dramatic change in body stiffness that occurs with age in *C. elegans* [14] suggests that epithelial cells are highly elastic and may need greater stimulus to fully active mechanotransduction pathways.

In addition to a lack of changes in actin-related genes, our RNA-seq analysis also did not show changes in UPR$^{MT}$ or other mitochondrial stress-related pathways. This was surprising considering the fragmentation of mitochondria observed in cells grown on stiff substrates and the activation of UPR$^{MT}$ pathways visible in other stiffness and integrin-related paradigms [17, 43]. Perhaps animals grown on stiffer substrates may not activate integrin signaling to a sufficient level to induce transcriptional changes but do increase mechanosignaling pathways enough to promote actin quality, which can have an impact on mitochondrial morphology. This is an important distinction as animals grown on stiffer substrates did not exhibit an increase in resilience to mitochondrial or oxidative stress, whereas animals with increased integrin signaling or cells grown on stiffer substrates in vitro did exhibit a significant increase in both mitochondrial and oxidative stress resilience [17, 43]. Moreover, animals with increased UPR$^{MT}$ signaling also exhibit an increase in resistance to paraquat [50], providing further evidence that animals grown on stiffer substrates do not activate a canonical UPR$^{MT}$ pathway. Important to note, however, is that increasing the agar content of growth medium has been shown to exacerbate muscle degeneration in a *C. elegans* model of muscular dystrophy, demonstrating that substrate stiffness does impact *C. elegans* physiology in multiple ways [20].

Although changes to the transcriptome were limited, a few lysosomal genes related to lipid metabolism, including lysosomal lipases, were identified. A previous study showed that downstream of UPR$^{ER}$ activation, increased lysosomal lipases can drive increased ER stress resilience and longevity [51]. This is an important consideration as animals grown on stiffer substrates also exhibit an increase in ER stress resilience, which is dependent on the UPR$^{ER}$ regulator, *xbp-1s*. While these data may suggest that the increase in lysosomal lipases and the associated increase in ER stress resilience may drive the lifespan extension found in these animals, we did not observe global changes to lysosome content. In addition, we also did not see any changes to whole animal lipid levels. Thus, further studies are required to determine whether the mild changes in lysosome-related genes identified in our study are sufficient to drive ER stress resilience and longevity phenotypes.

Even more interesting would be to determine whether there is any overlap between the ER and actin-related phenotypes. Indeed, a recent study has shown that promoting actin organization during aging by increased expression of the chromatin remodeling factor BET-1 has direct impact on ER stress resilience [22]. Moreover, the mammalian homologue of BET-1, BRD4, has been linked to pathological fibrosis in multiple tissues, including the cornea [52] and lung [53], likely through regulation of mechanosignaling. Thus, actin and ER regulation may have significant overlap, which can potentially be downstream of mechanotransduction pathways.

Overall, our study illustrated that whole animals grown on stiffer substrates do have general changes to cell biology, physiology, healthspan, and lifespan, although these effects are quite moderate. Although we can only speculate that these effects are due to increased agar stiffness inducing mechanical stress, several of these changes are consistent with other models of mechanotransduction, albeit not all pathways and changes seen in other systems were observed in our animals. While the phenotypes presented in this study may be mild and may not fully recapitulate what is expected for a mechanotransduction model, there is an important lesson here for *C. elegans* biologists. It is imperative that NGM agar plates not be stored long-term, as desiccation of plates can indirectly increase agar percentage relative to water content. Though this is only one facet of the many changes that can occur with desiccated plates, the change in relative agar-content can affect biological data, especially for phenotypes similar to

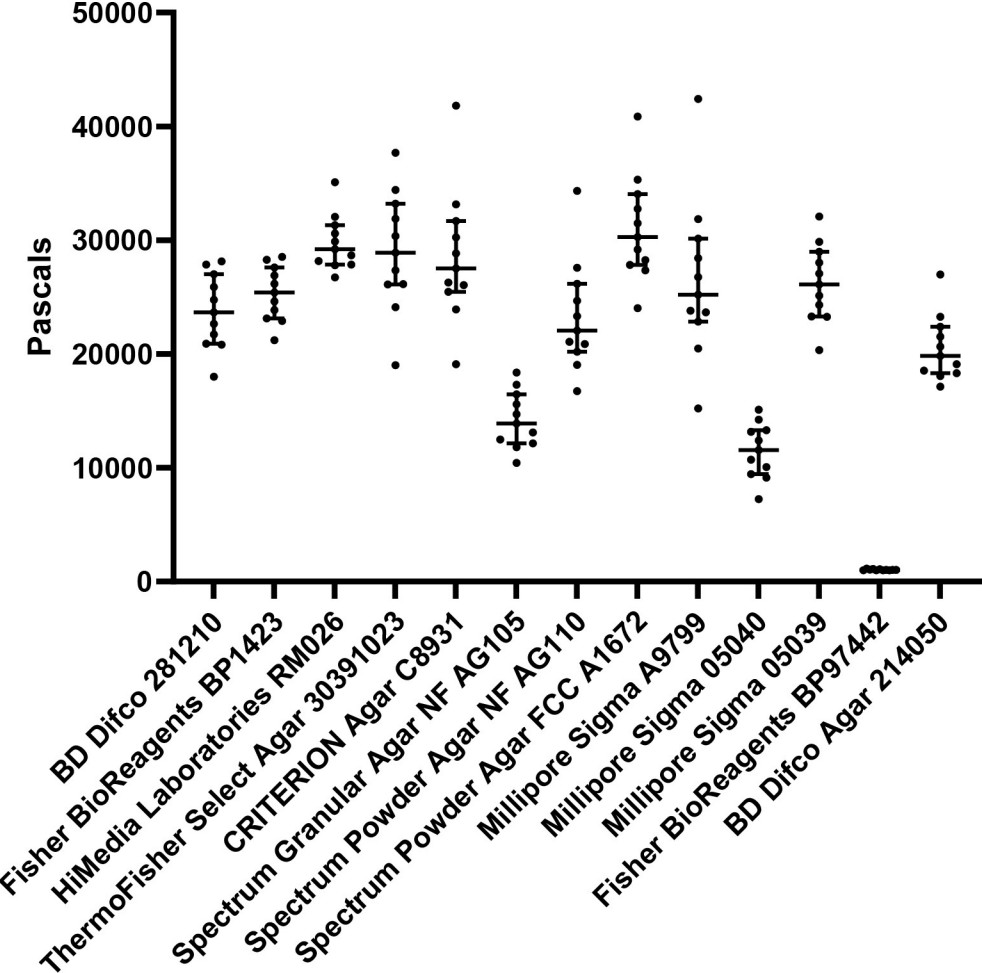

**Fig 5. Variable sources of agar formulations have significant differences in stiffness.** Standard 2% agar-based solid NGM plates were made as described in Materials and Methods using different sources of agar. Stiffness of each agar plate was measured using an oscillatory rheometer. Each dot represents a single technical replicate performed on an individual plate and lines represent median plus interquartile range. X-axis indicates brand and catalog number of each agar source.

those assayed in this study. Another very important consideration is to standardize an agar choice, as different agar sources can vary dramatically in stiffness (**Fig 5**). Again, stiffness is only one variable and there may be other differences between agar sources; however, the importance of maintaining consistency in reagents is clear. While our study exemplifies the importance of standardizing an agar choice for *C. elegans* biology, it is possible that this consideration is important for any study involving agar-based solid medium.

## Supporting information

**S1 Fig. 4% agar NGM plates are significantly stiffer than 2% agar NGM plates.** 2% and 4% agar-based solid NGM plates were made as described in Materials and Methods. Stiffness of each agar plate was measured using an oscillatory rheometer. Dots represents mean and lines represent standard deviation.
(TIF)

**S2 Fig. Growth on stiff substrates has no impact on lysosomes or lipid droplets. (A)** Representative max projection fluorescent images of lysosomes by visualization of LMP-1::GFP. Images were captured on a Leica Stellaris system using optimized z-slices. **(B)** Representative fluorescent images of lipid droplets by visualization of DHS-3::GFP. For A-B, animals were grown on empty vector (EV) RNAi bacteria from L1 and imaged at day 1 of adulthood. Scale bar is 10 μm. **(C)** Quantification of B. Arbitrary fluorescent units are integrated fluorescent densities as measured by imageJ Fiji. Data is representative of 2 independent replicates. (TIF)

**S3 Fig. Growth on stiff substrates increases ER stress resilience. (A)** N2 wild-type animals grown on empty vector (EV) or *xbp-1* RNAi bacteria on either control (2%) or stiff (4%) agar plates from L1. Animals were transferred onto the same RNAi and agar concentration plates containing 25 μg/mL tunicamycin (TM). Lifespans were scored every 2 days. Data is representative of 3 biological replicates, sample size is represented in the legend in parentheses, and statistical analysis is available in **S1 Table**. *** p < 0.0001 **(B)** N2 wild-type animals grown on EV RNAi bacteria on either control (2%) or stiff (4%) agar plates from L1. Animals were moved to plates containing 2.5 mM paraquat (PQ) on day 1 of adulthood and survival was scored every 2 days. Data is representative of 3 biological replicates, sample size is represented in the legend in parentheses, and statistical analysis is available in **S2 Table**. n.s. = not significant, p = 0.681. **(C)** Representative fluorescent images of day 1, 5, and 9 adult animals expressing *hsp-4p*::*GFP* grown on EV RNAi bacteria from L1. Animals were transferred onto plates containing 25 μg/mL tunicamycin 24 hours prior to imaging on L4, day 4, and day 8. **(D)** Quantification of (C) where arbitrary fluorescent units are integrated fluorescent intensity measurements using ImageJ Fiji, dots are independent replicates, and lines represent standard deviation. Data is representative of 2 independent replicates. ns = not significant, p > 0.05 comparing age-matched animals grown on 2% or 4% agar. (TIF)

**S4 Fig. Growth on stiff substrates does not impact ER or mitochondrial structure. (A)** Representative max projection fluorescent images of the endoplasmic reticulum in the intestine by visualization of HDEL::mRuby. Images were captured on a Leica Stellaris system using optimized z-slices. Animals were grown on empty vector (EV) RNAi bacteria from L1. Animals were moved onto 25 μg/mL tunicamycin containing plates at L4 and imaged at day 1 of adulthood. **(B)** Representative fluorescent images of body wall muscle mitochondria (*myo-3p*::*MLS*::*GFP*) are shown. Animals were grown on EV RNAi bacteria from L1. Single-slice images were captured on a Leica THUNDER Imager. Scale bar is 10 μm. **(C)** Representative max-projection fluorescent images of intestinal mitochondria (*gly-19p*::*MLS*::*GFP*) are shown. Animals were grown on EV RNAi bacteria from L1. Z-stack images were captured on a Leica THUNDER Imager using system-optimized z-slices. Scale bar is 10 μm. **(D)** Representative max-projection fluorescent images of hypodermal mitochondria (*col-19p*::*MLS*::*GFP*) are shown. Animals were grown on EV RNAi bacteria from L1. Z-stack images were captured on a Leica Stellaris using system-optimized z-slices. Scale bar is 10 μm. (TIF)

**S5 Fig. Growth on stiff substrates results in a mild increase in thermotolerance. (A-C)** N2 wild-type animals grown on empty vector (EV) RNAi bacteria on either control (2%) or stiff (4%) agar plates from L1. Animals were moved to 34˚C on day 1 of adulthood and survival was scored every 2 hours. All three biological replicates are shown to highlight differences in maximal thermotolerance, despite similarities in median thermotolerance in some replicates. Sample size is represented in the legend in parentheses and statistical analysis is available in

**S2 Table**. *** p < 0.0001 **(D)** 2% and 4% plates were placed in a 34˚C incubator and the temperature of the plates was measured using an infrared thermometer every 15 minutes. Each dot represents one plate and lines represent mean and standard deviation. ns = not significant, p > 0.05; ** = p < 0.01 using Mann Whitney testing.
(TIF)

**S1 Table. RNAseq analysis.**
(CSV)

**S2 Table. Statistics for survival curves.**
(XLSX)

**S3 Table.**
(DOCX)

## Acknowledgments

We thank all members of the Sanabria and Tharp labs who have assisted in the manuscript. In addition, we thank all mentors and members of the Gerontology Enriching MSTEM program, including Ms. Stanlee Gardner, Dr. Cristal Hill, Dr. Constanza Cortes, Dr. Sean Curran, Dr. Jennifer Ailshire, Dr. John Walsh, Mr. Bradford Barnes, and Ms. Maria Henke for all the assistance, guidance, and support provided to Maria Oorloff in carrying out the research study. Some strains were provided by the Caenorhabditis Genetics Center (CGC) and significant gene analysis was performed using Wormbase.

## Author Contributions

**Conceptualization:** Kevin M. Tharp, Gilberto Garcia, Ryo Higuchi-Sanabria.

**Data curation:** Maria Oorloff, Maxim Averbukh, Naibedya Dutta, Toni Castro Torres, Matthew Vega, Juri Kim, Andrew Bong, Aeowynn J. Coakley, Daniel Hicks, Jing Wang, Tiffany Wang, Sally Hoang, Gilberto Garcia, Ryo Higuchi-Sanabria.

**Formal analysis:** Maria Oorloff, Adam Hruby, Maxim Averbukh, Naibedya Dutta, Cray Minor, Kevin M. Tharp, Gilberto Garcia, Ryo Higuchi-Sanabria.

**Funding acquisition:** Ryo Higuchi-Sanabria.

**Investigation:** Maria Oorloff, Adam Hruby, Maxim Averbukh, Naibedya Dutta, Cray Minor, Toni Castro Torres, Juri Kim, Kevin M. Tharp, Gilberto Garcia, Ryo Higuchi-Sanabria.

**Methodology:** Maria Oorloff, Athena Alcala, Darius Moaddeli, Kevin M. Tharp, Gilberto Garcia, Ryo Higuchi-Sanabria.

**Project administration:** Gilberto Garcia, Ryo Higuchi-Sanabria.

**Resources:** Athena Alcala, Darius Moaddeli, Gilberto Garcia, Ryo Higuchi-Sanabria.

**Supervision:** Gilberto Garcia, Ryo Higuchi-Sanabria.

**Validation:** Maria Oorloff, Adam Hruby, Maxim Averbukh, Naibedya Dutta, Toni Castro Torres, Matthew Vega, Juri Kim, Andrew Bong, Aeowynn J. Coakley, Daniel Hicks, Jing Wang, Tiffany Wang, Sally Hoang, Kevin M. Tharp, Gilberto Garcia, Ryo Higuchi-Sanabria.

**Visualization:** Maria Oorloff, Adam Hruby, Maxim Averbukh, Naibedya Dutta, Toni Castro Torres, Matthew Vega, Kevin M. Tharp, Gilberto Garcia, Ryo Higuchi-Sanabria.

**Writing – original draft:** Maria Oorloff, Adam Hruby, Kevin M. Tharp, Gilberto Garcia, Ryo Higuchi-Sanabria.

**Writing – review & editing:** Maria Oorloff, Adam Hruby, Maxim Averbukh, Athena Alcala, Naibedya Dutta, Toni Castro Torres, Darius Moaddeli, Matthew Vega, Juri Kim, Andrew Bong, Aeowynn J. Coakley, Daniel Hicks, Jing Wang, Tiffany Wang, Sally Hoang, Kevin M. Tharp, Gilberto Garcia, Ryo Higuchi-Sanabria.

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
