## [Decision Letter · Decision Letter 0]

29 May 2024

PONE-D-24-14188Mechanical stress through growth on stiffer substrates impacts animal health and longevity in C. elegans.PLOS ONE

Dear Dr. Sanabria,

Thank you for submitting your manuscript to PLOS ONE. After careful consideration, we feel that it has merit but does not fully meet PLOS ONE’s publication criteria as it currently stands. Therefore, we invite you to submit a revised version of the manuscript that addresses the points raised during the review process.

We look forward to receiving your revised manuscript.

Kind regards,

Florian Rehfeldt

Academic Editor

PLOS ONE

Journal Requirements:

3. Please expand the acronym “CIRM” (as indicated in your financial disclosure) so that it states the name of your funders in full.

"A.H., M.A., and G.G. are supported by T32AG052374; M.O., T.C.T., and M.V. are supported by 1R25AG076400 from the National Institute on Aging; T.W. is supported by the CIRM COMPASS Award EDUC5-13853; and R.H.S. is supported by R00AG065200 from the National Institute on Aging, Larry L. Hillblom Foundation Grant 2022-A-010-SUP, and the Glenn Foundation for Medical Research and AFAR Grant for Junior Faculty Award. Some strains were provided by the CGC, which is funded by the NIH Office of Research Infrastructure Programs (P40 OD010440). Some gene analysis was performed using Wormbase, which is funded on a U41 grant HG002223."

"A.H., M.A., and G.G. are supported by T32AG052374; M.O., T.C.T., and M.V. are supported by 

1R25AG076400 from the National Institute on Aging; T.W. is supported by the CIRM 

COMPASS Award EDUC5-13853; and R.H.S. is supported by R00AG065200 from the National 

Institute on Aging, Larry L. Hillblom Foundation Grant 2022-A-010-SUP, and the Glenn 

Foundation for Medical Research and AFAR Grant for Junior Faculty Award. Some strains were 

provided by the CGC, which is funded by the NIH Office of Research Infrastructure Programs 

(P40 OD010440). Some gene analysis was performed using Wormbase, which is funded on a 

U41 grant HG002223."

"A.H., M.A., and G.G. are supported by T32AG052374; M.O., T.C.T., and M.V. are supported by 1R25AG076400 from the National Institute on Aging; T.W. is supported by the CIRM COMPASS Award EDUC5-13853; and R.H.S. is supported by R00AG065200 from the National Institute on Aging, Larry L. Hillblom Foundation Grant 2022-A-010-SUP, and the Glenn Foundation for Medical Research and AFAR Grant for Junior Faculty Award. Some strains were provided by the CGC, which is funded by the NIH Office of Research Infrastructure Programs (P40 OD010440). Some gene analysis was performed using Wormbase, which is funded on a U41 grant HG002223."

6. Thank you for stating the following in your Competing Interests section:  

"NO authors have competing interests"

Additional Editor Comments:

As indicated by Reviewer #1, it is imperative to add proper control experiments on how agar concentration might affect other parameters.

Indeed, as also mentioned by Reviewer #1, mechano-sensing phenomena of adherent cells are based on direct connections e.g. focal adhesions

for force transmission unlike C. elegans moving on the surface of substrates. This aspect and the authors' unsusubstantiated claim that a different stiffness of the agar substrate induces mechanical stress to C. elegans.

Within the manuscript they do not show any proof or evidence that this is actually the case.

closely brought this paper towards a plain "Reject" decision. You should be very careful in using the term mechanical stress, as the definition is force per area. Changing the agar concentration changes primarily the Young's modulus (i.e. stiffness) of the material.

Please take these points and all other issues raised by both referees very seriuosly into account when working on the major revisions for this manuscript.

Reviewers' comments:

Reviewer's Responses to Questions

**Comments to the Author**

1. Is the manuscript technically sound, and do the data support the conclusions?

Reviewer #1: Partly

Reviewer #2: Partly

2. Has the statistical analysis been performed appropriately and rigorously? 

Reviewer #1: I Don't Know

Reviewer #2: No

3. Have the authors made all data underlying the findings in their manuscript fully available?

Reviewer #1: Yes

Reviewer #2: Yes

4. Is the manuscript presented in an intelligible fashion and written in standard English?

Reviewer #1: Yes

Reviewer #2: Yes

5. Review Comments to the Author

Reviewer #1: This paper considers the hypothesis that growing animals on stiffer substrates will have an impact on their physiology by mechanotransduction pathways. The data and experiments are appropriately presented and the paper is clearly written, however, I think the way the experiments are framed isn't supported by the data. Specifically, changing the agar concentration from 2-4% will increase stiffness, but it may also change other aspects of the experiments that could have an impact on worm behaviour and physiology. For example, the plates may dry differently and worms are known to be sensitive to humidity. Different concentrations of agar could affect bacterial growth or physiology that in turn affects the worms. Indeed, even a subtle change on bacterial physiology could explain the small changes in worm phenotypes. Ruling this out may be difficult, but without much better controls, I see how the authors can conclude that the observed effects are mechanical.

Related to this, I don't share the authors shock at seeing only subtle effects of stiffness on worm physiology. Unlike adherent cells which bind to their substrates through integrins and pull on them, worms sit on top of agar in both the 2 and 4% conditions. Similarly, I wouldn't be shocked to see that people with thick carpets have very different physiology to people with thin carpets. And if they did, I wouldn't look for changes in mechanotrasduction related genes to explain any difference. What is the model for an impact on the worms? My expectation is that the forces the cells in the worm experience are dominated by muscle contraction not direct interaction with the substrate. If anything, because worms sink a little bit into the agar, I might imagine more mechanical stress in the softer condition (a bit like walking in deeper mud).

Given the difficulty of isolating stiffness from other properties of the system, the results and discussion should be framed around agar concentration differences rather than stiffness differences. Of course it's fine to speculate that the differences are due to stiffness, but without many more controls and a model of how stiffness could plausibly stress worms mechanically, it should be clearly marked as speculation.

Personally I would also remove the statements around being 'surprised' or 'shocked' that worms were 'wildly unaffected', but it's just a question of style.

Minor points:

-Does tunicamycin interact with agar? Is it possible some of it is sequestered by the higher agar concentration or that it’s diffusion is slowed so that there is a slight depletion at the agar surface where the worms are?

-For Fig 3B wouldn’t a plot of values be more useful than a heatmap? Also maybe easier to read if the genes are sorted by expression change?

-The authors state that “stiffer substrates can result in integrin-dependent remodeling of the mitochondria and actin cytoskeleton in both human cells [13] and C. elegans [31]", however these results did not show that stiffer substrates affect worm mitochondria, they show that integrin signalling in worms can affect mitochondria. The discussion of these results is fine elsewhere in the paper.

-The statement that "actin stability [...] is likely responsible for the mild increase in longevity" isn't sufficiently supported by the data. Many other things could be going on including mild calorie restriction because of differences in bacterial growth.

-From the discussion: "Perhaps animals grown on stiffer substrates may not activate integrin signaling to a sufficient level to induce transcriptional changes but do increase mechanosignaling pathways enough to promote actin quality". A direct connection here seems unlikely to me without some model for how stiffer substrates would activate a mechanosignaling pathway except perhaps in a sensory neuron.

-From the discussion: "It is imperative that NGM agar plates not be stored long-term, as desiccation of plates can indirectly increase agar precentage relative to water content. These plates will be stiffer and can change your biological data, especially for phenotypes similar to those we assayed in this study. Another very important consideration is to standardize an agar choice in the lab since multiple different agar sources have dramatically different stiffnesses". I agree with the advice, but I doubt stiffness is the main contributor and I don't think it's supported by the data presented here.

Reviewer #2: Mechanical signaling plays a pivotal role in many biological phenomena, including cell growth, differentiation, and aging. In their study, Oorloff et al. employ the C. elegans model to investigate the impact of external mechanical stress (2% vs. 4% agar gel) on aging, stress responses, and cytoskeletal proteins. The authors report that external mechanical stress modestly extends lifespan at the expense of reduced locomotor activity and reproductive capacity. Although several stress responses, as indicated by transcriptional markers, appeared unaffected, RNA sequencing revealed minimal gene expression changes. The authors suggest that actin dynamics may partially mediate the observed effects of mechanical stress on aging. However, due to the marginal nature of these effects, they conclude that mechanical stress has a limited impact on the studied phenotypes.

While this paper might capture the interest of the C. elegans research community, the core assertion that environmental mechanical stress significantly influences these outcomes lacks robust support, with deficiencies in biological context explanation, control usage, and data quantification. Several critical issues require attention:

Major concerns:

1. While using C. elegans to study the impact of mechanical stress on animal health and behavior is valid, the authors oversimplify the mechanistic implications of environmental and extracellular matrix-derived mechanical stresses. The introduction may lead readers to assume uniform effects of mechanical forces at both organismal and cellular levels. The discussion acknowledges the potential confounding effect of the C. elegans cuticle, but this issue warrants more extensive consideration in both the introduction and discussion sections.

2. The authors would like to use C. elegans as a model to study the effect of external mechanical stress on physiology. It is noteworthy that the body stiffness gradually becomes softer in aging C. elegans (PMID:32098962 ), which contradicts to examples in introduction. However, this piece of evidence suggests that mechanical forces might be an indicator of aging or even likely involve in aging. It is sad that this paper was not included and mentioned in the introduction and discussion. In fact, this work shows that body stiffness drastically changes over 20 folds along aging, suggesting that epithelial cells are highly elastic to maintain body integrity and probably have different sensitivities to external mechanical stress. It also raises an issue that whether 2-fold changes of agar can reveal biological alterations from the current setting.

3. Mechanical forces also influence the bacteria that serve as food for C. elegans (PMID: 36712798). The potential indirect effects of mechanical forces via changes in food source quality or abundance necessitate additional controls to distinguish these influences from direct interactions between the agar and C. elegans.

4. Despite claims that stress responses were generally unaffected, observed data from fluorescent images suggest increased stress responses in stiffer substrates under stress conditions. Quantitative data should be provided to substantiate this conclusion.

5. The claim that actin stability is influenced by agar stiffness is based solely on the abundance of actin, which does not directly reflect actin dynamics. Evidence supporting this claim or a revision of the statement to more accurately reflect the findings is needed.

Minor concerns:

1. The number of experiments conducted (N) and statistical analyses are missing in most figures. Consider relocating this information from figure legends directly into the figures for better clarity.

2. Allele names should be italicized to adhere to standard scientific formatting (in methods).

3. A dedicated statistical methods section is absent in the methodology description, which is crucial for reproducibility and validation of the results.

6. PLOS authors have the option to publish the peer review history of their article (what does this mean?). If published, this will include your full peer review and any attached files.

Reviewer #1: No

Reviewer #2: No

---

## [Author Response · Author response to Decision Letter 0]

17 Jun 2024

A complete response to reviewers has been attached as part of manuscript documents.

---

## [Decision Letter · Decision Letter 1]

10 Jul 2024

Growth on stiffer substrates impacts animal health and longevity in C. elegans.

PONE-D-24-14188R1

Dear Dr. Sanabria,

We’re pleased to inform you that your manuscript has been judged scientifically suitable for publication and will be formally accepted for publication once it meets all outstanding technical requirements.

Kind regards,

Florian Rehfeldt

Academic Editor

PLOS ONE

Additional Editor Comments (optional):

Reviewers' comments:

Reviewer's Responses to Questions

**Comments to the Author**

1. If the authors have adequately addressed your comments raised in a previous round of review and you feel that this manuscript is now acceptable for publication, you may indicate that here to bypass the “Comments to the Author” section, enter your conflict of interest statement in the “Confidential to Editor” section, and submit your "Accept" recommendation.

Reviewer #1: All comments have been addressed

Reviewer #2: All comments have been addressed

2. Is the manuscript technically sound, and do the data support the conclusions?

Reviewer #1: Yes

Reviewer #2: Yes

3. Has the statistical analysis been performed appropriately and rigorously? 

Reviewer #1: I Don't Know

Reviewer #2: Yes

4. Have the authors made all data underlying the findings in their manuscript fully available?

Reviewer #1: No

Reviewer #2: Yes

5. Is the manuscript presented in an intelligible fashion and written in standard English?

Reviewer #1: Yes

Reviewer #2: Yes

6. Review Comments to the Author

Reviewer #1: Just a note on data availability. The PLOS policy says that data points underlying estimates of summary statistics should be made available but from what I can see only the expression levels are given in the supplementary data but not the data underlying other figures.

Reviewer #2: The authors have addressed all my concerns. I thus have no more questions and support the publication.

7. PLOS authors have the option to publish the peer review history of their article (what does this mean?). If published, this will include your full peer review and any attached files.

Reviewer #1: No

Reviewer #2: No

---

## [Editor Report · Acceptance letter]

30 Jul 2024

PONE-D-24-14188R1 

PLOS ONE

Dear Dr. Sanabria, 

I'm pleased to inform you that your manuscript has been deemed suitable for publication in PLOS ONE. Congratulations! Your manuscript is now being handed over to our production team.

Kind regards, 

on behalf of

Dr. Florian Rehfeldt 

Academic Editor

PLOS ONE